# Length of Hospitalization and Mortality among Stroke Patients before and after the Implementation of a Specialized Unit: A Retrospective Cohort Study Using Real-World Data from One Reference Hospital in Southern Brazil

**DOI:** 10.3390/healthcare12080836

**Published:** 2024-04-15

**Authors:** Marcia Poll, Rodrigo Targa Martins, Fernando Anschau, Geraldo Pereira Jotz

**Affiliations:** 1Graduate Program in Health Sciences, Federal University of Health Sciences of Porto Alegre (UFCSPA), Porto Alegre 90050-170, RS, Brazil; 2Stroke Unit Coordination, Conceição Hospital Group, Porto Alegre 91350-200, RS, Brazil; 3Conceição Hospital Group, Department of Education and Research Coordination, Porto Alegre 91350-200, RS, Brazil

**Keywords:** stroke, stroke unit, length of hospitalization, hospital mortality, indicators

## Abstract

Stroke constitutes a significant global cause of mortality and disability. The implementation of stroke units influences hospital quality indicators, guiding care management. We aimed to compare hospital length of stay (LOS), in-hospital mortality, and post-discharge mortality between stroke patients admitted in the pre- and post-implementation periods of a stroke unit in a public hospital in southern Brazil. This retrospective cohort study used real-world data from one reference hospital, focusing on the intervention (stroke unit) and comparing it to the general ward (control). We analyzed the electronic medical records of 674 patients admitted from 2009 to 2012 in the general ward and 766 patients from 2013 to 2018 in the stroke unit. Admission to the stroke unit was associated with a 43% reduction in the likelihood of prolonged hospitalization. However, there was no significant difference in the risk of in-hospital mortality between the groups (Hazard ratio = 0.90; Interquartile range = 0.58 to 1.39). The incidence of death at three, six and twelve months post-discharge did not differ between the groups. Our study results indicate significant improvements in care processes for SU patients, including shorter LOS and better adherence to treatment protocols. However, our observations revealed no significant difference in mortality rates, either during hospitalization or after discharge, between the SU and GW groups. While SU implementation enhances efficiency in stroke care, further research is needed to explore long-term outcomes and optimize management strategies.

## 1. Introduction

Stroke is the second leading cause of death and a major contributor to global disability [1,2], with ischemic stroke accounting for approximately 85% of all cases [3,4]. Due to its high prevalence, therapeutic measures supported by robust scientific evidence, such as intravenous thrombolysis up to 4.5 h from symptom onset [5,6,7,8], endovascular treatment for proximal occlusion stroke [9,10,11,12], decompressive craniectomy for malignant edema resulting from extensive stroke [13,14], and the establishment of geographically delimited stroke units (SU) dedicated to stroke treatment [15], are now in place.

In recent decades, the Brazilian Ministry of Health (BMH) implemented public policies to improve stroke care. In 2012, the Emergency Care Network of Sistema Único de Saúde (SUS) introduced clinical protocols and therapeutic guidelines (CPTG), encompassing thrombolysis use [16] and the establishment of graded stroke units (types I, II, III) [17]. Type I SUs are reference hospitals that provide care for stroke patients, perform the procedure using thrombolytic drugs and have the minimal qualifications to treat the acute phase of stroke. Type II SUs have an acute stroke unit with at least five beds, provide thrombolytic agents, and ensure care within 72 h of admission in a multiprofessional fashion. These SUs provide physical and speech therapy, with a neurologist coordinating the patient’s treatment. Type III SUs have a 10-bed comprehensive stroke unit where treatment begins in the acute phase using a thrombolytic agent. Treatment extends to early rehabilitation, lasting up to 15 days of hospitalization, and includes occupational therapy, social work, and psychological care [17]. All Brazilian SU types are eligible for public funding from the Brazilian Ministry of Health, contingent upon delivering significant results monitored by quality-of-care indicators. The evidence supports specialized stroke care’s efficacy. Patients treated in SUs experience higher survival rates [15,18], independence [15], and a greater likelihood of returning home [15] compared to those treated in GWs. Moreover, the impact of SUs extends beyond the acute phase, as they aim to optimize functionality and enhance long-term quality of life [19].

In Brazil, the scarce literature related to the care of AIS victims presents contrasting findings regarding the benefits of SU implementation. In 2003, Cabral et al. did not find evidence of SU care superiority over GW regarding hospital length of stay (LOS) and morbidity, though their small sample size limited them from drawing conclusive results [20]. A decade later, Rocha et al. demonstrated that Brazilian SU care, while not impacting LOS, correlated with reduced in-hospital and 30-day mortality rates compared to GW care [21]. A recent systematic review analyzed the effectiveness of implementing AIS protocols in reducing care time, encompassing primary studies published between 2011 and 2020 [22]. Notably, among the included studies, none involved Brazilian patients.

In the context of neurological emergency care, Nossa Senhora da Conceição Hospital (HNSC), part of the Unified Health System (Sistema Único de Saúde, SUS), inaugurated its SU (type III) in May 2013. HNSC constitutes the largest tertiary hospital complex in the southern region of Brazil—the Conceição Hospital Group (GHC)—and annually serves as a referral for hospitalizing 46.1 thousand patients. The SU accommodates ten observation beds, five designated for acute ischemic stroke (AIS) care and the remaining for palliative care and rehabilitation. The SU’s healthcare multidisciplinary team includes neurologists, nurses, dietitians, physical therapists, occupational therapists, speech therapists, social workers, and psychologists. Moreover, neurology and multidisciplinary residents provide healthcare focused on SU patients; they follow a CPTG and initiate rehabilitation interventions immediately after events. Thus, this retrospective cohort study using real-world data from one reference hospital in southern Brazil aimed to compare the hospital LOS, in-hospital mortality, and death within three, six, and twelve months post-discharge in a sample of patients admitted for AIS pre- and post-SU implementation at HNSC.

## 2. Materials and Methods

### 2.1. Study Design and Ethical Aspects

This retrospective cohort study using real-world data from one reference hospital in southern Brazil focused on SU implementation (intervention under investigation) conducted at the GHC-HNSC in Porto Alegre, Brazil, from March 2009 to August 2018. The pre-intervention group (GW) (control) comprised patients admitted for AIS from March 2009 to December 2012, while the intervention group (SU) included patients admitted for AIS from November 2013 to August 2018. We excluded patients admitted between January and October 2013 because it represented the SU consolidation period.

The treatment of all AIS patients followed the stroke care flow, including risk assessment by the multidisciplinary team upon admission to the emergency unit. Patients within the reperfusion window or pre-screened by the mobile emergency service with suspected intracranial hemorrhage remained in the red zone, while cases outside the reperfusion window that were stable remained in the orange zone. All patients underwent cranial computed tomography and/or brain magnetic resonance imaging. AIS patients meeting thrombolysis criteria received priority for admission to SU, and the healthcare team directed those requiring endovascular treatment to a university hospital equipped with this technology. The inclusion of endovascular therapy in the SU only received approval in February 2021 through SCTIE/MS Ordinance No. 5/2021, following the publication of the Resilient study results [23]. 

Patients with hemorrhagic stroke (HS) were referred to the Cristo Redentor Hospital (HCR), also part of GHC, and were not eligible for the present study. Furthermore, all AIS patients received treatment under the same CPTG, including palliative care for severe cases, early rehabilitation, and decompressive craniectomy, guided by recommendations from the HCR neurosurgery team. Moreover, not all IS patients were admitted to the SU after its establishment, partly due to the prevalence of stroke in GHC’s covered area and the limited number of available beds, thus being directed to the neurology unit, internal medicine, or vascular surgery.

The Research Ethics Committee of the Conceição Hospital Group approved this study and waived the need for written informed consent from participants (Decision No. 3396327) according to national legislation and institutional requirements. Additionally, it received registration with the Research Commission of the Federal University of Health Sciences of Porto Alegre under No. 118/2019. 

### 2.2. Participants and Data Collection

The eligibility criteria included patients older than 18 years admitted to the HNSC emergency department with a primary clinical diagnosis of AIS, with it being their first event treated at this hospital, characterized by International Classification of Diseases ICD-10 as I63, I65, and I66 codes and respective subcategories by year of admission [24]. We reviewed patients’ medical records initially categorized under the ICD-10 codes I64 and I63.9, which correspond to unspecified strokes. Subsequently, we included these patients in the study following confirmation of acute AIS diagnosis. Exclusion criteria encompassed diagnoses mimicking AISs, transient ischemic attacks (TIA), or HSs. As we relied on secondary data, we individually reviewed and addressed missing or discrepant data whenever possible to minimize bias. We then excluded any remaining missing data from the analysis.

The research team, consisting of a trained nurse, two medical doctors, and a biomedical undergraduate student, obtained patient data from the electronic medical record system. These data included age, sex, home city, and comorbidities (vascular risk factors, preexisting morbidities, and behavioral characteristics), documented in medical records or registered as secondary ICD diagnoses.

For clinical evaluation, we utilized the following data collected during admission to the emergency room: the patient’s history as reported by both the patient and their relatives, encompassing a history of systolic pressure ≥140 mmHg, diastolic pressure ≥90 mmHg, and/or antihypertensive medication use. Additionally, we considered a fasting glycemia history ≥126 mg/dL and/or the use of antidiabetic agents. Furthermore, we took into account the use of lipid-lowering drugs or a history of hypertriglyceridemia and/or total cholesterol >200 mg/dL.

Preexisting health conditions of interest encompassed prior strokes with chronic lesions observed in CT scans and chronic kidney disease (changes in glomerular filtration rate, presence of parenchymal lesions for at least three months, or dialysis therapy). We grouped preexisting morbidities in the cardiac and respiratory systems accordingly. The cardiac system included morbidities such as acute myocardial infarction, congestive heart failure, atrial fibrillation, dilated cardiomyopathy, valvular stenosis, intra-atrial communication, and endocarditis. The respiratory system included morbidities such as pneumonia, chronic obstructive pulmonary disease, atelectasis, and acute lung edema. Behavioral characteristics of interest comprised active smoking or a history of smoking (≥20 cigarettes/day), current or a history of alcohol usage (21 and 14 units of alcohol consumption/week for men and women, respectively), or obesity (body mass index ≥30 kg/m^2^).

Researchers also collected indicators related to SU consolidation, encompassing door-to-needle time (DTNT) (time difference between the patient’s arrival at the emergency unit and the administration of intravenous thrombolytic therapy), antiplatelet therapy (antiplatelet agents and anticoagulation used in therapeutic dosage), deep vein thrombosis (DVT) prophylaxis (heparin or low-molecular-weight heparin administration), thrombolysis, SU admission and discharge dates, hospital discharges with prescriptions of antithrombotics, statins, and rehabilitation plans, and stroke-related death date. 

### 2.3. Outcomes of Interest

The primary outcomes for our study were those occurring within the hospital (LOS and death), while the secondary ones encompassed those after hospital discharge (mortality at three, six, and twelve months). We converted the hospital LOS into a categorical variable considering its observed median value in our sample, defining prolonged hospital LOS as values higher or equal to the sample median. Additionally, we assessed a hospital LOS-related outcome, a quality indicator proposed by the BMH, with a target result of less than nine days [25]. We followed up on the patients through medical records for 12 months to obtain information on mortality after hospital discharge.

### 2.4. Statistical Analysis

Descriptive statistics were utilized for data characterization, presenting non-parametric variables through medians and interquartile ranges (25P–75P). We described categorical variables using absolute and relative frequencies and tested quantitative variable distributions by employing the Kolmogorov–Smirnov test. The comparison variables between GW and SU groups for quantitative and categorical variables utilized the Mann–Whitney “U” and chi-square tests, respectively. 

We conducted a subgroup analysis for each outcome of interest (in-hospital and post-discharge mortality and LOS), aiming to assess which patient groups, if any, could benefit most from SU care. For this, we considered potential demographical and clinical outcome predictors. Demographic variables included sex, age group (adults <60 years versus older adults ≥60 years, under Brazilian law [26]), and geographical region of origin (Porto Alegre versus metropolitan area cities). For clinical characteristics, we created three composite variables based on the presence ‘+’ (at least one) or absence ‘−’ (none) of vascular risk factors (hypertension, diabetes mellitus, or dyslipidemia), comorbidities (previous stroke [ischemic, hemorrhagic, or TIA], cardiovascular disease, chronic kidney disease, chronic respiratory disease, or DVT and vasculopathy), and behavioral characteristics (smoking, alcohol consumption, or obesity).

We assessed the impact of SU implementation on prolonged hospital LOS (longer than the median value observed in our sample) by employing logistic regression. In addition, we performed Cox regression to evaluate the association between the intervention and in-hospital death. We developed both unadjusted and adjusted models. For the latter, we selected confounders based on *p*-values < 0.10 observed in the univariate analysis comparing the GW and SU groups. We did not consider any variable related to SU treatment to be confounders in the multivariable analysis since they are intrinsic to SU care.

The sample size calculation for this retrospective cohort considered an odds ratio (OR) of 0.44 for in-hospital mortality between AIS patients managed in the SU (6.9%) and those in GW (14.7%), as demonstrated in the study conducted by Rocha et al. [21], adopting a power of 80%, based on approximation with continuity correction, a significance level of 5%, and an additional 20% to account for adjustments in a multivariable model. Thus, using an online calculator (https://www.openepi.com/SampleSize/SSCohort.htm, accessed on 26 March 2024) we estimated a sample size of 779 patients. 

## 3. Results

This study included 674 (46.8%) and 766 (53.2%) patients from the GW and the SU, respectively. Table 1 presents a comparative analysis of demographic and clinical characteristics between the patient groups. The SU group had a higher proportion of women and individuals from Porto Alegre than the GW group. Patients in the GW group showed a higher frequency of hypertension, cardiovascular diseases, dyslipidemias, chronic kidney and respiratory diseases, and vasculopathy. However, the frequencies of diabetes, previous AIS or HS and TIA, obesity, smoking, and alcoholism showed no significant differences between groups.

Table 2 details the confirmatory variables related to SU consolidation. The SU group demonstrated a significant increase in patients receiving DVT prophylaxis and antiplatelet medications initiated within the first two days. Thrombolysis upon admission was more frequent in the SU group, and the time taken for thrombolysis was shorter compared to the GW group. Furthermore, at hospital discharge, SU patients showed an increase in receiving antithrombotic and statin prescriptions and rehabilitation plans (Table 2). 

The total patient sample exhibited a median hospital LOS of 11.2 days (interquartile range 7.2–18.3), while the GW group presented a median of 12.5 days (interquartile range 8.2–19.4), and the SU group a median of 10 days (interquartile range 6.2–16.2) (*p* < 0.0011). Patients in the GW group experienced a significantly prolonged hospital LOS compared to those in the SU group (Figure 1). The overall in-hospital death rate was 5.8% (5.0% and 6.8% for SU and GW patients, respectively). However, this difference was not statistically significant (Figure 1). Similarly, for secondary outcomes, no significant differences were observed between the groups regarding death incidences within three, six, and twelve months post-hospital discharge (Figure 1).

Table 3 and Table 4 present the subgroup univariate analysis comparison of outcome variables. Younger patients experienced a significant decrease in in-hospital mortality in the SU (OR = 0.96, 95% CI 0.94–0.99). The analysis found no significant difference in post-discharge mortality for any subgroup of SU patients. However, patients managed in the SU across all examined subgroups demonstrated a notable reduction in hospital LOS (all OR < 0.7).

Logistic regression revealed a 43% reduction in the chance of prolonged hospital LOS for the SU group after adjusting for age, sex, home city, and comorbidities (Table 5). The odds of SU patients meeting the hospital LOS quality indicator (<nine days) was 1.87 (95% CI: 1.49–2.33) times higher than those of GW patients. Conversely, Cox regression analysis did not reveal a significant association between in-hospital death and SU implementation (Table 5). We did not conduct multivariate analyses exploring the impact of SU implementation on mortality within the first 12 months post-hospital discharge, due to the lack of association demonstrated in univariate analysis.

## 4. Discussion

This retrospective cohort study using real-world data from one reference hospital in southern Brazil compared the hospital LOS, in-hospital mortality, and death incidences within three, six, and twelve months after discharge among AIS patients admitted to HNSC pre- and post-SU implementation. The aim was to investigate the potential benefits of SU implementation. The findings revealed that implementing SU was associated with a decrease in the likelihood of patients being hospitalized for more than 11 days, resulting in a 43% reduction in the probability of a prolonged hospital LOS. However, SU implementation did not independently predict in-hospital mortality, nor did it exhibit an association with death rates within the specified post-discharge timeframes of three, six, and twelve months.

The assessment of the SU consolidation indicators in this study underscored its effectiveness within the HNSC, with increases in thrombolysis frequency, early DVT prophylaxis, initiation of antiplatelet medications within 24 h, and the number of patients with antithrombotics, statins, and personalized rehabilitation plans at hospital discharge. These results confirm the consolidation of specialized care for stroke patients, aligning with findings from previous studies that compared indicators before and after SU implementation. Other studies conducted in various primary and integrated stroke centers demonstrated that providing care in an SU increases the frequency of thrombolysis and significantly reduces DTNT [27,28]. In our study, admission to the SU significantly reduced DTNT compared to the GW group, with median times of 79 min and 122 min, respectively. Furthermore, the number of thrombolysis patients within <60 min post-SU implementation increased from 7 to 25. Although modest, these findings parallel those of a study conducted in Botucatu (SP), which also identified a small number of patients undergoing thrombolytic treatments with door-to-needle times <60 min [29]. This timeframe is consistent with data from European registries, which reported a mean time of 70 min [30]. Despite the increase in thrombolysis frequency post-SU implementation (6.1% to 10.8%), it remained relatively low, underscoring the ongoing need to strengthen the stroke survival chain [29]. However, this is similar to the reality in northern and southern European countries, where only 7.3% of all AIS patients receive thrombolysis, reflecting an unequal geographical SU distribution [31].

Following SU implementation, a notable rise occurred in DVT prophylaxis frequency and antiplatelet medication usage upon admission and discharge. These enhancements persisted despite being already prevalent practices, with over 90% of patients using antithrombotics and over 80% using statins before SU implementation. Similarly, other studies reported high rates of prophylaxis and antiplatelet medication prescription upon discharge, with DVT prophylaxis performed in over 85% and the antiplatelet drugs prescribed for more than 70% of cases [32]. In the UK, over 80% of at-risk patients of recurrent stroke receive antiplatelet treatment [33]. Global surveys indicate that hospitals administer antiplatelet agents and statins to 50–60% of patients upon discharge [34]. This underscores the importance of adhering to guidelines for prophylactic antithrombotic therapy to reduce the risk of recurrent vascular events, thereby enhancing care quality for AIS and TIA patients while mitigating associated costs [32,33,34,35,36].

In this study, SU implementation led to an approximately 1.8% increase in the frequency of discharging patients with a rehabilitation plan. Research consistently shows that providing rehabilitation plans upon hospital discharge for stroke patients significantly enhances their quality of life, reduces the likelihood of sequelae, and decreases readmission rates, along with in-hospital and late clinical complications [30,32,36,37]. A study conducted in Bahia highlighted the limited availability of comprehensive stroke care in most Brazilian hospitals, with only secondary and prophylactic treatments provided due to the unavailability of thrombolysis and thrombectomy, despite their regulation by the SUS [38]. Additionally, patients admitted for AIS to HNSC in pre- and post-SU implementation differed regarding various comorbidities and sex distribution. These differences may have introduced confounding variables that impacted the association between SU implementation and the outcomes of interest. To address this, we considered these variables as confounders in a multivariate model constructed to explore their association with prolonged hospital LOS.

To evaluate the effectiveness of SU implementation, this study compared outcomes related to prolonged hospital LOS, in-hospital mortality, and late mortality among patients admitted for stroke pre- and post-its implementation. The results demonstrated a significant difference of approximately 2.5 days in the median hospital LOS between the two groups, with the SU associated with a 43% reduction in the likelihood of patients remaining hospitalized for more than 11 days. Furthermore, considering the quality indicator set by the BMH for hospital LOS < nine days [25], patients in the SU group were 1.87 times more likely to meet this indicator than those in the GW group. However, a prior study conducted in a public hospital in Joinville reported a mean hospital LOS of 11 days in the SU group and 12.6 days in the GW group, with no significant difference [20]. The limited study power due to small sample sizes (35 patients in the SU group and 39 in the GW group) likely explains these results. Similarly, a study conducted in a hospital in São Paulo with 729 AIS patients admitted to the GW and 344 patients admitted to the SU showed no difference in the hospital LOS between the groups, with a median of 10.2 days [21].

The present study did not show a significant association between in-hospital death or death within three, six, and twelve months after hospital discharge and the implementation of the SU. A study at Joinville Hospital (SC) also found no difference in death incidence among patients admitted to the GW and the SU [20]. In contrast, a study at São Paulo Hospital (SP) revealed a 33% reduction in the relative risk of death within 30 days in an analysis adjusted for confounders [21]. The incidence of death within 30 days was 20.9% and 14.2% in the GW and SU groups, respectively [21], whereas in our sample, the incidence of in-hospital death decreased from 6.8% in the GW group to 5.0% in the SU group. This low death incidence may explain the lack of association between SU implementation and this outcome. In addition, the absence of long-term survival impacts observed in our SU compared to the pre-implementation period, despite improvements in acute care, may be attributed to stroke patients’ access to rehabilitation professionals (not assessed in our study). Magalhães et al.’s research [39] revealed suboptimal access to rehabilitation professionals among patient groups (before and during the COVID-19 pandemic) one month post-discharge from a Brazilian public SU. This finding underscores compromised care comprehensiveness for AIS, stemming from inadequacies in stroke care infrastructure, a shortage of specialized professionals, and a misalignment between the rehabilitation system and patient needs in the Brazilian public healthcare system.

Aligned with the Brazilian Ministry of Health, this study demonstrates good adequacy levels for quality indicators of in-hospital care, affirming the ongoing relevance of continuous monitoring. However, Brazil still faces a lack of investment in SUs [29]. Despite this, Brazil, a middle-income country in the Americas, stands out due to its standardized care consolidated through national health policies in stroke treatment outlined by the WHO. The public healthcare system provides comprehensive care and free universal coverage, similar to health policies adopted by France, a high-income country, in stroke treatment. Brazil exhibits lower in-hospital mortality rates and shorter LOS than France [40]. In this context, the present study highlights successful strategies for in-hospital care within the stroke care network, which can support increased investment in SUs for low- and middle-income countries. 

In Brazil, the implementation of integrated stroke care strategies, assessed through quality indicators, offers valuable insights for comparable contexts in low- and middle-income countries [41]. Our positive intrahospital indicators among 1440 patients represent robust benchmarks, particularly for neighboring Latin American nations. The progression of stroke care in Brazil reflects the hurdles encountered by middle-income countries in advancing stroke prevention, treatment, and rehabilitation. This experience holds relevance for informing and enhancing stroke care approaches in similar healthcare settings worldwide [41].

This study is not without limitations. Firstly, its retrospective data collection prevented access to complete data for all patients of interest. Specifically, it hindered the assessment of stroke severity upon admission using the National Institutes of Health Stroke Scale (NIHSS) [42], an essential predictor of stroke outcome, which neurologists commonly do not apply in many developing countries, including our study setting. Secondly, we did not conduct an etiological classification of AISs according to the Trial of ORG 10172 in Acute Stroke Treatment (TOAST) [43], which guides therapeutic decisions. Thirdly, AIS patients admitted to the HNSC pre- and post-SU implementation exhibited differences in various comorbidities and sex. These discrepancies may have introduced confounding variables affecting the association between SU implementation and outcomes of interest. To mitigate this, we included these variables as confounders in a multivariate model constructed to explore the association with prolonged hospital LOS.

The interpretation of our study findings should be cautious, as we compared in-hospital and twelve-month post-discharge outcomes between pre-and post-implementation of the SU in two cohorts of AIS patients over different periods, potentially influenced by temporal changes unrelated to SU implementation not identified as confounding factors in the results. However, this reflects real-world clinical practice. In addition, post-discharge data collection occurred exclusively through the consultation of patients’ electronic medical records, potentially resulting in the loss of study patients readmitted to another hospital. However, given that HNSC is a referral hospital, there was a higher likelihood of patients getting readmitted to the same facility, which may mitigate concerns regarding the number of losses. Finally, the low number of deaths in the study may have affected its ability to identify associations with mortality.

Future studies should assess the impact of SU implementation on the incidence of in-hospital complications, a factor not investigated in this study due to clinical and sociodemographic disparities between the groups. Furthermore, exploring the effect of SU implementation on hospital costs and examining other quality indicators related to specialized care for these patients could enhance understanding of the benefits associated with SU. With a robust evidence base, SU could be implemented in more hospitals, leading to substantial improvements in the quality of life for patients affected by this disease.

## 5. Conclusions

In conclusion, the results of this retrospective cohort study using real-world data from one reference hospital indicate significant improvements in care processes for patients in the SU, including shorter LOS and better adherence to treatment protocols. However, our observations revealed no significant difference in mortality rates, either during hospitalization or after discharge, between the SU and GW groups. In the context of SUs in Brazil and low- and middle-income countries, there is a need for additional evidence, including strategies to support long-term care and recovery in the community.

## Figures and Tables

**Figure 1 healthcare-12-00836-f001:**
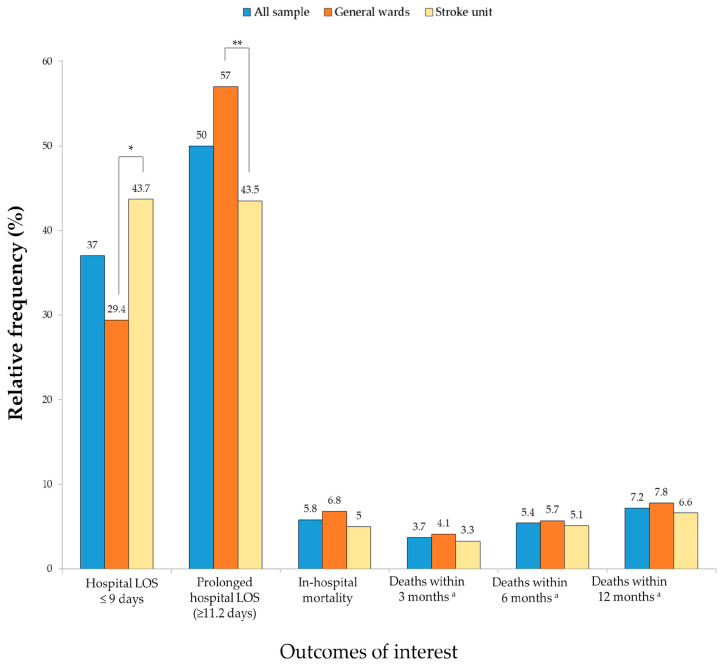
Comparison of the primary and secondary outcomes among stroke patients in the general ward and stroke unit groups (n = 1440). Abbreviations: LOS, length of stay. * *p* < 0.001 by Mann–Whitney Test, ** *p* < 0.001 by chi-square test. ^a^ n = 1356.

**Table 1 healthcare-12-00836-t001:** Demographic and previous clinical characteristics of patients admitted due to ischemic stroke in the general ward (2009 to 2012) and stroke unit (2013 to 2018) of the Conceição Hospital Group, Porto Alegre, Brazil. (n = 1440).

Variables	Total Sample (n = 1440)	GW Group (n = 674)	SU Group (n = 766)	*p*-Value
Demographic data				
Female, n (%)	731 (50.8)	315 (46.7)	416 (54.3)	0.005 ^1^
Age, years	66 (56.5–75.5)	66 (56–75)	66 (58–76.3)	0.084 ^1^
Home city				
Porto Alegre, n (%)	911 (63.3)	199 (59.2)	512 (66.8)	0.010 ^2^
Metropolitan Region of Porto Alegre, n (%)	410 (28.5)	211 (31.3)	199 (26)	
Countryside of RS, n (%)	119 (8.3)	64 (9.5)	55 (7.2)	
Vascular risk factors, comorbidities, and behavioral characteristics				
Systemic arterial hypertension, n (%)	1013 (70.3)	497 (73.7)	516 (67.4)	0.008 ^2^
Diabetes mellitus, n (%)	473 (32.8)	236 (35)	237 (30.9)	0.100 ^2^
Dyslipidemias, n (%)	181 (12.6)	114 (16.9)	67 (8.7)	<0.001 ^2^
Previous stroke				
Ischemic or hemorrhagic, n (%)	269 (19)	138 (20.8)	131 (17.4)	0.210 ^2^
TIA, n (%)	28 (2.0)	11 (1.7)	17 (2.3)	
Cardiovascular disease, n (%)	240 (16.7)	137 (20.3)	103 (13.4)	<0.001 ^2^
Chronic kidney disease, n (%)	52 (3.6)	35 (5.2)	17 (2.2)	0.003 ^2^
Chronic respiratory disease, n (%)	47 (3.3)	32 (4.7)	15 (2.0)	0.003 ^2^
DVT + vasculopathy, n (%)	30 (2.1)	21 (3.1)	9 (1.2)	0.010 ^2^
Smoking, n (%)	376 (26)	171 (25.4)	204 (26.6)	0.586 ^2^
Alcohol consumption, n (%)	114 (7.9)	61 (9.1)	53 (6.9)	0.135 ^2^
Obesity, n (%)	89 (6.2)	34 (5)	55 (7.2)	0.093 ^2^

Data presented as medians (interquartile ranges) or absolute and relative frequencies; Abbreviations: GW, general ward; SU, stroke unit; RS, Rio Grande do Sul; TIA, transient ischemic attack; DVT, deep vein thrombosis; ^1^ Mann-Whitney Test, ^2^ Chi-Square Test.

**Table 2 healthcare-12-00836-t002:** Comparison of confirmatory variables related to the consolidation of the SU: treatment of ischemic stroke among patients admitted to the general ward (2009 to 2012) and stroke unit (2013 to 2019) of the Conceição Hospital Group, Porto Alegre, Brazil. (n = 1440).

Variables	Total Sample (n = 1440)	GW Group (n = 674)	SU Group (n = 766)	*p*-Value
Thrombolysis upon admission	124 (8.6)	41 (6.1)	83 (10.8)	0.002 ^1^
Time to thrombolysis (DTNT) (minutes)	86.0 (56.0–126.0)	122.0 (83.5–177.3)	79.0 (56.0–111.5)	0.002 ^2^
DTNT <60 min	32 (26)	7 (17.1)	25 (30.5)	0.167 ^1^
DVT Prophylaxis initiated up to the 2nd day ^a^	1340 (93.1)	610 (90.5)	730 (95.3)	<0.001 ^1^
Administration of antiplatelet medications ^b^	1331 (92.4)	618 (91.7)	713 (93.1)	<0.001 ^1^
Hospital discharge				
with antithrombotics ^c^	1251 (87.1)	575 (85.4)	676 (88.5)	<0.001 ^1^
with statins ^c^	1242 (86.4)	550 (81.7)	692 (90.6)	<0.001 ^1^
with rehabilitation plans ^d^	703 (50.1)	224 (35.1)	479 (62.7)	<0.001 ^1^

Data presented as medians (interquartile ranges) or absolute and relative frequencies; Abbreviations: GW, general ward; SU, stroke unit; DTNT, door-to-needle-time; DVT, deep vein thrombosis. ^a^ n = 1408; ^b^ n = 1391; ^c^ n = 1437; ^d^ n = 1403. ^1^ Mann–Whitney Test, ^2^ chi-square test.

**Table 3 healthcare-12-00836-t003:** Subgroup analyses according to in-hospital outcome measures among stroke patients admitted to the general ward (2009 to 2012) and stroke unit (2013 to 2019) of the Conceição Hospital Group, Porto Alegre, Brazil. (n = 1440).

Subgroup	Mortality (n = 84)			Prolonged Length of Stay (n = 721)		
	GW (%)	SU (%)	OR (95% CI)	GW (%)	SU (%)	OR (95% CI)
Male	6.4	3.4	0.52 (0.25–1.06)	52.6	42.0	0.65 (0.48–0.88)
Female	7.3	6.3	0.85 (0.47–1.51)	26.3	20.4	0.49 (0.36–0.66)
Age < 60	3.9	0	0.96 (0.94–0.99)	50.0	39.1	0.64 (0.44–0.93)
Age ≥ 60	8.3	7.0	0.82 (0.51–1.32)	61	45.4	0.53 (0.41–0.69)
Porto Alegre	6.0	5.3	0.87 (0.5–1.53)	55.6	43.4	0.61 (0.47–0.8)
MA cities	8.3	4.1	0.47 (0.21–1.10)	56.8	44.1	0.60 (0.41–0.87)
Vascular RF −	18.7	12.0	0.59 (0.32–1.10)	65.7	52.7	0.58 (0.37–0.93)
Vascular RF +	3.9	2.7	0.70 (0.36–1.35)	55.2	40.7	0.56 (0.44–0.71)
Behavioral −	9.2	6.0	0.78 (0.41–1.48)	58.4	46	0.61 (0.47–0.78)
Behavioral +	1.5	2.8	1.92 (0.49–7.50)	54.6	38.7	0.53 (0.36–0.76)
Comorbidity −	5.6	4.8	0.86 (0.6–1.58)	50	39.8	0.66 (0.50–0.87)
Comorbidity +	8.2	5.2	0.61 (0.32–1.18)	65.4	49.8	0.53 (0.38–0.73)

Abbreviations: OR, odds ratio; 95% CI, 95% Confidence Interval; MA cities, Metropolitan area cities.

**Table 4 healthcare-12-00836-t004:** Subgroup analyses according to post-discharge outcome measures among stroke patients admitted to the general ward (2009 to 2012) and stroke unit (2013 to 2019) of the Conceição Hospital Group, Porto Alegre, Brazil. (n = 1440).

Subgroups	Mortality Post-Discharge								
	Within 3 Months (n = 50)			Within 6 Months (n = 73)			Within 12 Months (n = 97)		
	GW (%)	SU (%)	OR (95% CI)	GW (%)	SU (%)	OR (95% CI)	GW (%)	SU (%)	OR (95% CI)
Male	3.3	1.4	0.42 (0.15–1.20)	4.2	3.4	0.81 (0.38–1.78)	5.0	4.3	0.85 (0.42–1.71)
Female	4.4	4.6	1.03 (0.51–2.09)	6.7	6.0	0.90 (0.49–1.63)	9.8	7.9	0.79 (0.47–1.31)
Age < 60	1.7	0.9	0.52 (0.1–2.9)	2.2	2.3	1.05 (0.3–3.66)	3.0	3.2	1.05 (0.36–3.03)
Age ≥ 60	5.0	4.0	0.81 (0.44–1.47)	7.0	5.9	0.82 (0.51–1.32)	9.5	7.5	0.78 (0.5–1.22)
Porto Alegre	4.5	2.9	0.64 (0.321–1.29)	6.5	5.1	0.77 (0.44–1.34)	9.3	6.6	0.70 (0.43–1.13)
MA cities	3.1	3.2	1.04 (0.36–3.02)	3.9	4.1	1.04 (0.41–2.68)	4.8	5.5	1.14 (0.49–2.65)
Vascular RF −	2.2	2.7	1.12 (0.29–5.20)	5.2	5.4	1.04 (0.39–2.81)	8.2	6.5	0.78 (0.33–1.83)
Vascular RF +	4.3	3.3	0.76 (0.41–1.41)	5.4	4.6	0.86 (0.50–1.47)	7.0	6.2	0.87 (0.54–1.40)
Behavioral −	4.5	3.5	0.64 (0.39–1.03)	6.4	5.7	0.88 (0.52–1.45)	9.0	7.4	0.81 (0.52–1.29)
Behavioral +	2.4	2.4	0.97 (0.29–3.23)	2.9	3.2	1.10 (0.37–3.17)	3.4	4.0	1.16 (0.43–3.11)
Comorbidity −	3.7	2.1	0.57 (0.25–1.31)	4.8	3.6	0.74 (0.37–1.47)	6.2	4.4	0.70 (0.38–1.30)
Comorbidity +	4.1	4.8	1.19 (0.55–2.57)	6.0	6.9	1.16 (0.61–2.22)	8.5	9.3	1.10 (0.63–1.93)

Abbreviations: OR, odds ratio; 95% CI, 95% Confidence Interval; MA cities, Metropolitan area cities.

**Table 5 healthcare-12-00836-t005:** Impact of the implementation of the SU on the length of hospitalization and hospital mortality: multivariate analysis.

Independent Variable	Prolonged Hospital LOS (>11.2 Days) ^1^ [OR (95% CI)]		In-Hospital Mortality ^2^ [HR (95% CI)]	
	Unadjusted	Adjusted ^a^	Unadjusted	Adjusted ^a^
Stroke Unit	0.56 (0.47–0.71)	0.57 (0.48–0.73)	1.0 (0.67–1.60)	1.1 (0.71–1.7)

Abbreviations: LOS, length of stay; OR, odds ratio; 95% CI, 95% Confidence Interval; HR, hazard ratio; ^1^ Logistic regression, ^2^ Cox regression; ^a^ Adjusted for age ≥60, vascular risk factors, behavioral characteristics, and comorbidities.

## Data Availability

Anonymized data not published within this article will be made available by request from any qualified investigator. Data are not publicly available due to ethical reasons. Further enquiries can be directed to the corresponding author.

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
