# Peer review of "Length of Hospitalization and Mortality among Stroke Patients before and after the Implementation of a Specialized Unit: A Retrospective Cohort Study Using Real-World Data from One Reference Hospital in Southern Brazil"

_healthcare, 2024, doi:10.3390/healthcare12080836_

Round 1

Reviewer 1 Report

Comments and Suggestions for Authors

Dear Authors,

Thank you for  scientific report of your cohort study.

The outcome variable should be separated by demographic variables (Table 1). One of the goals of the cohort study is to investigate the role of the influencing variables on the outcome. Please also address the point in the article. For example, which age group, which risk factor has the most mortality.

 Best regards,

Author Response

Dear Reviewer

Answer: Thank you for the suggestion to improve our manuscript. Following your suggestion, we opted to perform a subgroup analysis to evaluate which patient groups, if any, benefit the most from SU care. We described it in the Statistical analysis section, presenting its results in Tables 3 and 4, and the Results section text.

The modifications suggested were highlighted in red in the revised version of the manuscript. 

 Best regards,

Authors

Reviewer 2 Report

Comments and Suggestions for Authors

This is well-written in a way that a reader who is not a physician can understand. A minor recommendation for clarity:

Line 241 states: "In our study, admission to the SU demonstrated significantly decreased DTNT compared to those in the GW group, with a median time of 79 minutes and 122 minutes, respectively."  However, in the table, DTNT for the GW is "7" and the DTNT for the SU is 25. it's unclear what units the 7 and the 25 represent, please revise either the text to provide more information and explain the 7 and 25, or revise the Table to report the same units and numbers of information, I'm not sure how to interpret the 7 and 25 if this is medians of frequency counts of people.

Author Response

Dear Reviewer,

Thank you for your recommendation for clarity.

We have aligned the information provided in the text for clarity in the revised version of the manuscript.

The modifications suggested were highlighted in red in the revised version of the manuscript. 

 Best regards,

Authors

Reviewer 3 Report

Comments and Suggestions for Authors

The Length of Hospitalization and Mortality Among Stroke Patients Before and After the Implementation of a Specialized Unit study by the authors was quite valuable.

The abstract's conclusion section can be written better.

The introduction is quite brief and there are not many references cited in the introduction. It's expandable.

You can refer to more current and relevant references .
How the sample size determined?

There aren't any figures. Why?

Why the authors used Mann-Whitney "U" and chi-square tests??
What was the validation method?

More current and relevant references are needed for compare and discuss in the discussion section.

The conclusion section could be written more better. It doesn't adequately support the results.

In my opinion, Informed Consent form is needed.

Author Response

Reviewer #3

Comments and Suggestions for Authors

The Length of Hospitalization and Mortality Among Stroke Patients Before and After the Implementation of a Specialized Unit study by the authors was quite valuable.

The abstract's conclusion section can be written better.

Answer: Thank you for your feedback. We have modified the conclusion section of the abstract to enhance its effectiveness in the revised version of the manuscript.

The introduction is quite brief and there are not many references cited in the introduction. It's expandable.You can refer to more current and relevant references .

Answer: Thank you for your valuable feedback regarding our manuscript. We appreciate your suggestion to enhance the introduction by incorporating more references. We have considered your advice and have included several scientifically relevant references from the past five years in the respective section. We believe these additions will strengthen the comprehensiveness and relevance of our work.

How the sample size determined?

Answer: We appreciate your contribution. We clarified this by rewriting the sample size calculation, considering a Brazilian reference in the methods section.

There aren't any figures. Why?

Answer: We included a figure (Figure 1) in the results section to illustrate the primary and secondary outcomes between the stroke patients in the general ward and stroke unit groups.

Why the authors used Mann-Whitney "U" and chi-square tests??
What was the validation method?

Answer: We compared demographic and clinical characteristics between patient groups using Mann-Whitney U and chi-square tests. We choose these statistical tests due to their appropriateness for comparing continuous variables (such as age) and categorical variables (such as gender, city of origin, and medical conditions) between two independent groups.

We employed the Mann-Whitney U test to analyze continuous variables, such as age, for non-parametrical data distribution. On the other hand, we applied the chi-square test to assess the association between categorical variables, such as gender, city of origin, and presence of medical conditions (hypertension, cardiovascular diseases, dyslipidemias, etc.), across the two patient groups. This test determines whether there is a significant difference in the distribution of categorical variables between groups.

More current and relevant references are needed for compare and discuss in the discussion section.

Answer: Thank you for your feedback. We acknowledge the need for more current and relevant references in the discussion section, incorporating additional recent studies (published within the last five years) to strengthen our discussion section and facilitate comparisons with the latest findings in the field.

The conclusion section could be written more better. It doesn't adequately support the results.

Answer: Thank you for your insightful observation. We appreciate your attention to detail and constructive feedback. In response to your comment, we rewrited the conclusion section in the revised version of the manuscript.

In my opinion, Informed Consent form is needed.

Answer: Thank you for raising this concern. We acknowledge your concern and have addressed and clarified this information in the Informed Consent Statement section of the revised manuscript. It is noted that both the institutional ethics committee (attached file) and national legislation, RESOLUTION No. 466, OF 12 DECEMBER 2012, (https://conselho.saude.gov.br/resolucoes/2012/466_english.pdf ) waive the requirement for the signature of the Informed Consent form in studies utilizing the methodology employed in this investigation.

The modifications suggested were highlighted in red in the revised version of the manuscript. 

 Best regards,

Authors

Reviewer 4 Report

Comments and Suggestions for Authors

The manuscript presents an investigation into the effects of a stroke unit's introduction at a public hospital in southern Brazil, focusing on outcomes such as hospital stay duration, mortality rates during hospitalization, and mortality rates after discharge. This retrospective cohort study contrasts patient outcomes before and after the stroke unit's establishment. However, the analysis encounters several limitations that undermine its novelty and methodological rigor. First, the comparison involves cohorts from different time periods, which may introduce confounding variables not accounted for in the study design. Additionally, the manuscript appears to cover ground previously explored in the literature, as evidenced by its substantial overlap with the findings and methodologies of two earlier studies (DOI: 10.1590/s0004-282x2003000200006, DOI: 10.1590/0004-282X20130120). This repetition raises questions about the original contribution of the current study to the existing body of research on stroke unit efficacy.

Comments on the Quality of English Language

Minor editing of English language required

Author Response

Reviewer #4

Comments and Suggestions for Authors

The manuscript presents an investigation into the effects of a stroke unit's introduction at a public hospital in southern Brazil, focusing on outcomes such as hospital stay duration, mortality rates during hospitalization, and mortality rates after discharge. This retrospective cohort study contrasts patient outcomes before and after the stroke unit's establishment. However, the analysis encounters several limitations that undermine its novelty and methodological rigor. First, the comparison involves cohorts from different time periods, which may introduce confounding variables not accounted for in the study design. Additionally, the manuscript appears to cover ground previously explored in the literature, as evidenced by its substantial overlap with the findings and methodologies of two earlier studies (DOI: 10.1590/s0004-282x2003000200006, DOI: 10.1590/0004-282X20130120). This repetition raises questions about the original contribution of the current study to the existing body of research on stroke unit efficacy.

Answer: We appreciate your thoughtful consideration. Indeed, our study faces several methodological limitations. However, in the revised version of the manuscript, we detail its relevance in the introduction section. Similarly, we address the limitations regarding cohorts from different periods in the discussion section.

Comments on the Quality of English Language

Minor editing of English language required

Answer: In this revised version of the manuscript, we are grateful for the invaluable and voluntary grammatical review conducted by an American neurologist. Following this review, we are confident that the manuscript adheres to the grammatical English writing rules.

The modifications suggested were highlighted in red in the revised version of the manuscript. 

 Best regards,

Authors